# High-Pressure Processing for Cold Brew Coffee: Safety and Quality Assessment under Refrigerated and Ambient Storage

**DOI:** 10.3390/foods12234231

**Published:** 2023-11-23

**Authors:** Berta Polanco-Estibález, Rodrigo García-Santa-Cruz, Rui P. Queirós, Vinicio Serment-Moreno, Mario González-Angulo, Carole Tonello-Samson, Maria D. Rivero-Pérez

**Affiliations:** 1Hiperbaric SA, Polígono Industrial Villalonquéjar, Calle Condado de Treviño, 09001 Burgos, Spain; b.polanco@hiperbaric.com (B.P.-E.); r.queiros@hiperbaric.com (R.P.Q.); c.tonello@hiperbaric.com (C.T.-S.); 2Department of Biotechnology and Food Science, Faculty of Sciences, University of Burgos, Plaza de Misael Bañuelos, 09001 Burgos, Spaindrivero@ubu.es (M.D.R.-P.); 3Hiperbaric USA Corp., 2250 NW 84th Ave., Suite 101, Doral, FL 33122, USA

**Keywords:** high-pressure processing, cold brew coffee, *Escherichia coli* O157:H7, *Listeria monocytogenes*, *Salmonella enterica*

## Abstract

Cold brew coffee (CBC) has gained in popularity due to its distinct sensory experience. However, CBC can pose a risk for bacterial pathogens if not stored properly. High-Pressure Processing (HPP) is a nonthermal technology that can improve the safety of CBC while maintaining its quality. In this study, CBC made from ground roasted coffee grains was processed at 600 MPa for 3 min and stored at 4 or 23 °C for 90 days. The microbiological quality indicators remained stable throughout the study period. Physicochemical and quality parameters, such as pH, total dissolved solids, titratable acidity, color, total phenolic compounds and antioxidant activity, were not significantly affected by HPP. Both unprocessed and HPP CBC samples showed changes in pH, titratable acidity and color stability after 60 days at 23 °C. Unprocessed CBC samples spiked with *Escherichia coli* O157:H7, *Listeria monocytogenes* and *Salmonella enterica* showed decreased counts, but the pathogens were still detectable after 60 days at 4 °C and after 90 days at 23 °C. HPP achieved a >6-log_10_ reduction in the species tested, with non-detectable levels for at least 90 days at both storage temperatures. These findings suggest that HPP can effectively control vegetative pathogens and spoilage microorganisms in CBC while preserving its quality attributes.

## 1. Introduction

Coffee is one of the most widely consumed beverages in the world due to its aroma, pleasant taste and stimulating properties. In recent years, cold brew coffee (CBC) has gained more recognition among consumers. The brewing process for CBC involves immersing ground coffee in cold water (4–25 °C) for a prolonged period of time (12–24 h) followed by the removal of the solids. CBC develops a unique flavor profile, and tends to be less bitter and acidic than coffee brewed with hot water [1,2]. Compared to hot-brewed coffee, CBC also has a lower concentration of total phenolic compounds (TPCs) and a lower total antioxidant capacity (TAC), especially when light- and medium-roasted coffee is used [3]. This is attributed to the reduced capacity of cold water to extract poorly soluble and polar substances with antioxidant properties. However, the concentrations of caffeine and chlorogenic acid in hot and cold brews are similar [3,4].

Coffee brews are considered a poor substrate for microbial development, but research suggests that pathogenic bacteria may survive in the beverage for several weeks [5,6]. Additionally, some fungal species have the ability to grow and potentially produce mycotoxins, which drastically reduces the shelf-life of the final product and compromises its food safety [7]. In the particular case of CBC, these biological hazards may emerge as a consequence of long infusion times under refrigeration or even at room temperature, cross-contamination during production (either by personnel or contaminated food contact surfaces) and the lack of preservation steps after brewing and packaging. Particularly, Lopez (2020) reported that *Escherichia coli* O157:H7, *Listeria monocytogenes* and *Salmonella* spp. May be viable in CBC for 7 to 28 days. A study conducted by the Canadian Food Inspection Agency (CFIA) monitored microbial quality indicators (aerobic plate counts, coliforms) and pathogens (*E. coli* O157:H7, *Salmonella* spp.) in cold brew coffee by collecting 59 commercial samples of the beverage across Canada over one year [8]. Although no pathogens were detected, 25% of the samples screened presented >100 CFU/g for aerobic plate counts. Therefore, the implementation of strategies for the control of spoilage and pathogenic microorganisms is encouraged to ensure food safety and shelf-life extension of CBC.

High-Pressure Processing (HPP) is a nonthermal food preservation technology that uses hydrostatic pressure to inactivate microorganisms with a minimal impact on the sensory, nutritional and functional properties of food products [9]. The process uses water at cold or ambient temperature (4–25 °C) to create a pressure (up to 600 MPa) that is typically maintained for 1–5 min on already packaged products, or in-bulk on liquid products before the packaging step. In the beverage sector, HPP has been crucial in the commercialization of heat-sensitive juices made from fruits, vegetables and blends, smoothies, soups, plant-based dairy alternatives, or cold infusions like coffee or tea [9]. In this regard, some companies that take advantage of the potential benefits that HPP can bring to CBC are listed in the Appendix A.

HPP has the potential to ensure food safety as documented in a survey conducted by the CFIA. The study includes data of more than 1200 samples of commercial HPP juices sampled over one year in Canada. Microbial analyses indicated the absence of pathogenic non-spore-forming bacteria (*L. monocytogenes*, *E. coli* O157:H7, *Salmonella* spp.), viruses (hepatitis A, norovirus) and parasites (*Cryptosporidium parvum*, *Toxoplasma gondii*) [10]. This is consistent with other published research on HPP juices [11]. In addition to pathogen control, HPP is used to extend the shelf-life of food and beverages. In the particular case of CBC, Bellumori et al. (2021) [12] reported that HPP (608 MPa, 360 s) led to non-detectable levels of microbiological quality indicators (aerobic plate counts, yeasts and molds) through 120 days. The results were comparable to those obtained through heat pasteurization at 65 °C for 30 min. The same authors reported that the characteristic flavor of coffee infusions is also preserved via HPP since no significant differences were identified among 17 volatile compounds studied in non-treated and HPP-treated CBC samples. Other studies suggest that heat processing negatively affects the odor-active compounds of coffee drinks [13].

While the scientific literature assessing the high-pressure processing inactivation of *E. coli* O157:H7, *L. monocytogenes* and *S. enterica* in fruit juices is plentiful, there are no publicly available reports dealing with the control of the three major foodborne pathogens on CBC by means of HPP. Hence, the objective of this work is to assess the potential of HPP to control pathogens in CBC while increasing its shelf-life and maintaining its physiochemical and quality attributes under refrigeration or at room temperature.

## 2. Materials and Methods

### 2.1. Coffee Brewing

Infusion was prepared by placing a commercial blend of medium-roasted Arabica and Robusta ground coffee varieties (Hacendado, Spain) in a hemp filter bag (7% *w*/*v*). The bag was immersed for 17 h at 4 °C in mineral water (AGUADOY: TDS at 180 °C 226 mg/L; HCO_3_^−^ 184 mg/L; SO_4_^2−^ 5.8 mg/L; Cl^−^ 13.7 mg/L; NO_3_^−^ 13.3 mg/L; Ca^2+^ 27 mg/L; Mg ^2+^ 8.6 mg/L; Na^+^ 35.8 mg/L; K^+^ 1.4 mg/mL). After the infusion, the bag containing the solid ground coffee was removed and 30 mL aliquots of the brew were dispensed in transparent PET bottles (Sunbox, Spain).

### 2.2. Microbial Analyses

#### 2.2.1. Pathogen Inoculation

Five-strain cocktails of *E. coli* O157:H7, *L. monocytogenes* and *S. enterica* were created for the challenge test. Pathogenic species were selected based on the requirements for high-pressure processing validation [14]. Whereas *E. coli* O157:H7 and *Salmonella* spp. Have been associated with outbreaks related to fresh juice consumption, *L. monocytogenes* is of particular concern in ready-to-eat products because of its psychrotrophic nature. Strains used on each cocktail (Table 1) were selected based on their pressure resistance and adaptation phenotypes according to González-Angulo et al. (2021) [15]. Isolates were cultured from 20% (*v*/*v*) glycerol stocks (−80 °C) on tryptic soy agar plates with 0.6% (*w*/*v*) yeast extract (Oxoid, UK) at 37 °C for 24 h. Individual colonies were transferred to 10 mL of tryptic soy broth (Condalab, Spain) and incubated at 37 °C for 24 h to obtain fully active cultures.

For each individual species, equivalent volumes of each overnight-grown strain were combined in a 50 mL centrifuge tube to create three strain cocktails. The cocktails were washed through centrifugation at 12,900× *g* for 10 min at 4 °C in a 5810R Eppendorf centrifuge (Eppendorf, Germany). The supernatant was discarded and the cell pellet was resuspended in phosphate-buffered saline (pH 7.2) (Oxoid, UK). Then, each 30 mL CBC sample was inoculated with 300 µL of the appropriate bacterial cocktail to obtain an approximate cell concentration of 10^7^ CFU/mL. Half of the spiked samples were processed via HPP (see Section 2.3), whereas the other half remained unprocessed. Each set of processed and unprocessed samples was further divided for storage at 4 °C or 23 °C for 90 days. Triplicate samples were analyzed on each sampling point on days 0, 1, 7, 14, 28, 60 and 90.

Agar Listeria according to Ottaviani and Agosti (Scharlab, Spain) was used for the enumeration of *L. monocytogenes*, Sorbitol MacConkey Agar (Oxoid, UK) for *E. coli* O157:H7 and Xylose-Lysine-Desoxycholate Agar (Biolife, Italy) for *S. enterica*. Plates were incubated at 35 ± 2 °C for 24 h before colony enumeration. Microbiological counts were expressed as 10-base logarithmic colony-forming units (CFUs) per milliliter of coffee (log_10_ CFU/mL).

#### 2.2.2. Spoilage Microorganisms

In parallel, uninoculated CBC samples were likewise divided into two sets: one was processed through HPP (see Section 2.3) and one remained unprocessed. Each of these sets was again divided for storage at either 4 °C or 23 °C for 90 days. Figure 1 shows the distribution of samples for this experiment.

Typical food spoilage microorganisms were studied on days 0, 7, 14, 28, 60 and 90. Total mesophilic bacteria (TMB), yeasts and molds (YMs) and *Enterobacteriaceae* were analyzed as spoilage indicators by plating 1 mL (for TMB and *Enterobacteriaceae*) or 100 µL (for YM) of appropriately diluted CBC samples on Plate Count Agar (Oxoid, UK), Sabouraud Dextroxe Chloramphenicol agar (Oxoid, UK) and Violet Red Bile Glucose agar (Oxoid, UK), respectively. Each determination was carried out in triplicate throughout the experiment, and agar plates were incubated at 30 ± 1 °C for 48 h for TMB, 25 ± 1 °C for 72 h for YM, and 37 °C ± 1 °C for 24 h for *Enterobacteriaceae*. The same sets of uninoculated samples were used to determine the physicochemical characteristics of the CBC (see Section 2.4).

### 2.3. High-Pressure Processing (HPP)

High-pressure processing was carried out at 600 MPa for 3 min at ~18 °C in a Hiperbaric 135 L unit (Hiperbaric, Spain). The maximum operating pressure is 600 MPa, with a pressurization rate of around 255 MPa/min and instantaneous depressurization (<3 s). The temperature increase due to adiabatic heating was estimated at 3 °C/100 MPa during pressurization, which yielded a maximum temperature of ~36 °C under pressure. These processing conditions were selected because they are the standard used in the HPP beverage industry. The goal was to replicate the conditions used in an industrial setting. After HPP, processed samples were stored at either 4 °C or room temperature (23 °C).

### 2.4. Physicochemical Characterization of the Coffee Samples

All physicochemical determinations were conducted in triplicate for each condition (unprocessed and high-pressure-processed samples) and storage temperature (4 °C or 23 °C) throughout the experiment.

#### 2.4.1. Total Dissolved Solids (TDSs)

The TDSs were measured following the methodology proposed by Moreno et al. (2015) [16] and used by others for CBC characterization [1,4]. The relationship between the concentration of total soluble solids (X_s_) and the Brix degrees (°Brix) is defined as follows: X_s_ = 0.0087 × °Brix. Determination of Brix degrees was performed using an HI 96801 refractometer (Hanna Instruments, Nusfalau, Romania) at 20 °C.

#### 2.4.2. pH and Titratable Acidity (TA)

The pH value of the samples was measured at 25 °C with a calibrated glass electrode pH meter (micro pH 2001, Crison Instruments, Barcelona, Spain). TA was determined through titration of 25 mL of CBC with 0.1 mol/L NaOH solution to an endpoint pH of 6.5. The results are expressed in milligrams of chlorogenic acid per liter of coffee (mg CGA/L) (INCOTEC, 2004).

#### 2.4.3. Color Measurement

Color parameters *a** (redness), *b** (yellowness) and *L** (lightness) were measured using a Konica Minolta CM-2600d (Osaka, Japan) reflectance spectrophotometer with D65 illuminant and 10° standard observer angle. The total color difference (Δ*E*) was calculated as follows taking as reference the color parameters $\;L_{0}^{*}$, $a_{0}^{*}$ and $b_{0}^{*}$ of unprocessed coffee samples at day 0: $∆E^{*}={\left[{\left(L^{*}-L_{0}^{*}\right)}^{2}+{\left(a^{*}-a_{0}^{*}\right)}^{2}+{\left(b^{*}-b_{0}^{*}\right)}^{2}\right]}^{1/2}$.

#### 2.4.4. Total Polyphenolic Content (TPC)

TPC was evaluated using the Folin–Ciocalteu colorimetric method using gallic acid as a standard [17]. A volume of 20 μL of the Folin–Ciocalteau reagent was mixed with 20 μL of a 1:10 dilution of CBC. After 5 min of incubation at room temperature, 400 μL of a 75 g/L sodium carbonate solution was added, and the volume was then brought to 1 mL with Milli-Q water. The absorbance at 750 nm was measured after a second incubation in the dark at room temperature for 1 h. A standard curve of gallic acid was prepared, and the results are expressed as gallic acid equivalents (μg GAE/mL).

#### 2.4.5. Total Antioxidant Capacity (TAC)

Two different methodologies (ABTS and FRAP methods) were used for TAC determination. The ABTS method is based on the discoloration that occurs when the radical cation ABTS^●+^ is reduced to ABTS [18]. The assay consisted of 980 μL of ABTS^●+^ solution and 20 μL of appropriately diluted CBC. The absorbance at 734 nm was measured after 20 min of reaction. The results are expressed in millimolar of Trolox equivalent (mM Trolox/mL). The FRAP method is based on the increase in absorbance at 593 nm linked to the formation of 2,4,6-Tris(2-pyridyl)-S-triazine (TPTZ)−Fe^2+^ complexes in the presence of a reducing agent [19]. The reactive mixture was prepared by combining 25 mL of a sodium acetate buffer solution (0.3 M, pH 3.6), 2.5 mL of TPTZ (10 mM), 2.5 mL of FeCl_3_ (20 mM) and 3 mL of Milli-Q water. Then, 30 μL of appropriately diluted CBC was added to 970 μL of the reactive mixture and incubated at 37 °C for 30 min. The results are expressed as mM FeSO_4_·7H_2_O/mL.

### 2.5. Statistical Analysis

Results are expressed as the average of three replicates ± standard deviation. One-way analysis of variance (ANOVA) was used to identify significant differences between different processing and storage conditions within a sampling day, and between sampling days for the same processing and storage condition. The least significant difference (LSD) test was applied to determine the statistical significance between various groups. A minimum significance level of *p* ≤ 0.05 was considered. The Statgraphics Centurion v.19 software (Statpoint Technologies, Inc., Warranton, VA, USA) was used.

## 3. Results and Discussion

### 3.1. Physicochemical Characteristics

Total dissolved solids (TDSs) are related to the “body” of coffee, a parameter used by brewers to establish the strength and general flavor characteristics of the infusion [20,21,22]. This parameter increases with the degree of roasting and brewing temperature [4]. According to the Brewing Control Charts of the Specialty Coffee Association (SCA) used to balance the coffee brew flavor profile, the extraction yield (EY) should be 18–22%, with 0.79–1.38% TDSs [23]. Recently, Frost, Ristenpart and Guinard (2020) [22] from the UC Davis Coffee Center updated this chart correlating the TDS and EY with the sensory profile of drip-brewed coffee. Our findings show that unprocessed CBC samples presented an initial TDS of 1.4%, a value that was not affected by HPP. This indicator ranged between 1.4 and 1.6% throughout storage, regardless of the processing and storage temperature (Figure 2A). These findings are aligned with Lopane (2018) [24], who reported an average of 1.5% TDSs in unprocessed CBC with few variations during 42 days of storage at 7 °C. Similarly, Cordoba et al. (2019) [1] reported TDS values ranging from 0.7 to 2.0% on untreated CBC samples, depending on the gridding and contact time between coffee and water.

Both pH and titratable acidity (TA) are responsible for the perceived acidity in the coffee brew. The pH measures the concentration of hydronium ions, while TA is the measurement of all acidic protons in a sample, including non-dissociated protons. The initial pH of CBC samples was 5.88 ± 0.02, which is within the range of commercially available cold-brewed coffee [6]. The pH of unprocessed and high-pressure-processed CBC decreased at similar rates (up to 0.24 pH units) throughout the 90 days of storage at 4 °C, whereas a 0.7 pH unit drop was observed at room temperature (Figure 2B). An increase in hydrogen ion concentration is related to the quality loss of coffee brews as it affects the acid–base equilibria and the partition of volatile compounds in the vapor phase. In particular, a pH decrease has been attributed to the hydrolysis of carbohydrates and/or melanoidins [25]. However, pH values of CBC stored under refrigeration and room temperature would fall within the acceptance threshold of 4.80 considered by Rosa, Barbanti and Lerici, (1990) [26].

The initial TA of unprocessed CBC was 755 ± 73 mg CGA/L and was not affected by HPP (Figure 2C). Similarly to pH, differences in TA seem to be attributed to storage temperature rather than to processing. TA increased from 496 to 567 mg CGA/L in the CBC samples stored at 4 °C, compared to 1512 to 1724 mg CGA/L in the CBC samples stored at room temperature (Figure 2C). As expected, these results reveal a high correlation between the pH and TA in all conditions, in agreement with So et al. (2014) [27], who reported a faster pH decline and TA increase in CBC stored at room temperature than in brews kept under refrigeration. The increase in TA and the reduction in pH have been attributed to several factors, including the hydrolysis of the quinic/chlorogenic acid lactones formed during the roasting process, the hydrolysis of low-molecular-weight esters, and the degradation of chlorogenic acids into their corresponding hydroxycinnamic acids (e.g., caffeic acid, quinic acid) [28]. 

### 3.2. Shelf-Life Assessment

#### 3.2.1. Microbiological Quality

The initial microbial load of the unprocessed CBC was low (1.3 ± 0.1 log_10_ CFU/mL TMB, <1 log_10_ CFU/mL YM and <1 CFU/mL of *Enterobacteriaceae*). All microbiological indicators displayed counts below 1.3 log_10_ CFU/mL over 90 days, regardless of processing conditions and storage temperature (see Appendix A).

These results are in agreement with Lopane (2018) [24], who reported microbial counts below the detection limit (<1 CFU/mL) in untreated CBC over 42 days at 7 °C. López Parra et al. (2021) [29] also reported no microbial growth in high-pressure-processed CBC (600 MPa; 3 min) stored under refrigeration (4 °C) or room temperatures for 270 days. Nonetheless, the risk of microbial growth increases when brewing is performed at ambient temperature compared to refrigeration. In this regard, Bellumori et al. (2021) [12] reported initial TMB counts of 3 log_10_ CFU/mL, whereas no YMs were detected (<5 CFU/mL) in CBC brewed at 20 °C. The concentration of TMB and YM in unprocessed CBC samples reached 7 and 4 log_10_ CFU/mL, respectively, after 7 days of storage at room temperature. Interestingly, the authors reported that processing CBC at 608 MPa for 7 min served to extend the shelf-life up to 120 days, maintaining microbial indicators below the detection limit (<5 CFU/mL). This goes in agreement with the results presented in this study and evidences the potential of HPP to extend the shelf-life of CBC.

#### 3.2.2. Physicochemical Quality: Color, Total Polyphenol Content and Total Antioxidant Capacity

Some studies claim that the shelf life of coffee brews is not limited by microbial stability, but by the deterioration of sensory attributes instead [24]. The *L*a*b** color components of CBC did not significantly vary, because of processing or storage temperature, but changed through storage time (Table 2). As expected, the *L** component value was close to 0 at all times, which corresponds to the typical black color of coffee. The *a** component (+ red/− green) moved from reddish to greenish values. The *b** component (+ yellow/− blue) always showed positive values, corresponding to a yellowish color. The total color change of CBC was not affected by HPP on day 0 (∆*E* < 1), and remained stable for 60 days of storage at 4 °C and 23 °C. Beyond day 60, the ∆*E* value exceeded the limit in which a standard observer would notice a clear color difference compared to unprocessed samples on day 0 (Δ*E* > 3.5) [30] . This could be related to the above-mentioned reactions responsible for the increase in acidity. Specifically, nonvolatile compounds such as melanoidins or chlorogenic acids are responsible for the color change in coffee [24]. Additional sensory attributes, including flavor, were not formally evaluated. Nevertheless, a general assessment indicates that HPP did not considerably alter the flavor profile of CBC, consistent with the typical impact observed in beverages subjected to this treatment. These observations align with the findings of Bellumori et al. (2021) [12], who reported a substantial resemblance between untreated and HPP CBC. However, it is noteworthy that the scientific literature on the influence of HPP on the sensory characteristics of CBC is limited. Existing data primarily focus on the HPP extraction of CBC, with a scarcity of information regarding the application of pressure as a means to enhance the shelf-life and safety of the beverage post extraction [20,21,22].

The TPC ranged between 2000 and 2300 µg GAE/mL through 90 days of storage with no clear influence of processing conditions or temperature (Figure 3). Most of the differences observed were not significant (*p* >0.05), which suggests a good stability for these compounds in CBC. The results are similar to those described in other studies. López Parra et al. (2021) [29] reported no significant differences in the total and individual phenolic compounds in CBC subjected to HPP (600 MPa for 3 min) and stored at 4 °C for 270 days. Regarding individual phenols, the authors identified the presence of hydroxycinnamic acids, whereas isoflavones and anthocyanins were not detected. Bilge (2020) [31] found a TPC concentration in the range of 3000 to 4000 μg GAE/mL for CBC brewed at room temperature for 12 h. Similarly, So et al. (2014) [27] reported 2572–2951 µg GAE/mL in unprocessed CBC, with no significant differences in the concentrations among samples stored at 4 °C or 20 °C for 8 weeks.

In addition to TPC, it must be taken into consideration that CBC is rich in other bioactive molecules with high antioxidant capacity such as melanoidins, caffeine and certain volatile components. The total antioxidant capacity (TAC) was assessed using the ABTS and FRAP assays. As can be seen in Figure 4A, all samples maintained the antioxidant capacity against the ABTS radical, regardless of processing conditions or storage temperature during the first 28 days (i.e., around 11 mM Trolox equivalents). This is within the range reported in other studies [4,32,33]. However, a progressive rise in antioxidant capacity was observed from this time onward, with an increase between 63 and 72% by the end of the 90 days of storage. This same phenomenon was observed on UHT-preserved coffee brews, in which the scavenging activity was measured using the DPPH method [34,35]. Anese and Nicoli (2003) [25] attributed these changes to non-oxidative polymerization reactions of melanoidins or melanoidins precursors, which could account for the color variation previously discussed in the present study (Table 2). Coffee extraction procedures and/or storage conditions could favor the formation of Maillard products (such as phenol-type intermediates) that can increase the overall radical scavenging properties of the coffee brew [25]. On the other hand, the total antioxidant capacity determined using the FRAP assay suggested that neither processing conditions nor storage time and temperature had a significant effect on the antioxidant capacity of CBC (31–41 mM FeSO_4_·7H_2_O, Figure 4B). This stability could be associated with that observed in the concentration of TPC (Figure 3).

### 3.3. HPP Pathogen Inactivation

The initial concentration of *E. coli* O157:H7, *L. monocytogenes* and *S. enterica* on inoculated CBC samples ranged from 7.00 ± 0.02 to 7.43 ± 0.07 log_10_ CFU/mL. Processing CBC at 600 MPa for 3 min reduced the counts of the three pathogenic species below the detection limit (<1 log_10_ CFU/mL) and exceeded a 6-log_10_ reduction. Additionally, this reduction was sustained for 90 days, regardless of the storage temperature (Figure 5). Other authors reported that HPP consistently inactivates the three main pathogens on fruit and vegetable juices [36] or even on other low-acid (pH > 4.6) plant-based beverages [37].

No pathogen growth was observed on unprocessed CBC samples, but the three studied species remained viable for at least 60 days at 4 °C (>2 log_10_ CFU/mL) and 90 days at 23 °C (2 to 3 log_10_ CFU/mL). Overall, the cell concentration of *E. coli* O157:H7, *L. monocytogenes* and *S. enterica* did not change during the first 14 days, with the exception of *S. enterica* at 4 °C, which decreased by 2 log_10_ CFU/mL. Cell counts markedly decreased from day 30 onward, especially on CBC samples at 4 °C inoculated with *L. monocytogenes* and *S. enterica*. However, a 5-log_10_ reduction could only be observed after 60 days of storage at 4 °C. Interestingly, *S. enterica* was the only species that reached a 5-log_10_ reduction at 23 °C after 90 days, whereas *E. coli* O157:H7 and *L. monocytogenes* still displayed counts above 3 log_10_ CFU/mL. 

Similar findings were described by Acharya and Nummer (2022) [6] on twenty different varieties of CBC (pH 4.97 to 6.14) inoculated with *L. monocytogenes* (10^5^ CFU/mL) and stored under refrigeration (4 °C) for 60 days. The authors observed a decrease in *L. monocytogenes* counts over time, although some CBC varieties displayed counts between 0.43 and 3.78 log_10_ CFU/mL after 60 days. Another study reported that *Salmonella* spp., *L. monocytogenes* and *E. coli* could be still detected on inoculated CBC after 7, 10 and 14 days of storage at 4 °C, respectively [5].

The absence of microbial growth and the steady decrease in pathogens in CBC and hot infusions have been associated with the lack of essential nutrients for metabolism and/or the presence of naturally occurring compounds with antimicrobial properties in coffee brews. Compounds with potential antimicrobial effects include caffeine, trigonelline and phenolics like caffeic acid and chlorogenic acid, among others [38,39,40,41].

## 4. Conclusions

In this study, the characteristics of cold brew coffee (CBC) were examined, with a focus on physicochemical properties, shelf-life assessment and pathogen inactivation. Total dissolved solids, which indicate coffee strength, remained stable at approximately 1.4–1.6% in unprocessed CBC samples during storage. Whereas the pH decreased, titratable acidity increased over time, particularly at room temperature. This decline in pH and rise in titratable acidity were attributed to factors like the hydrolysis of compounds. Microbiological quality was consistently low, with microbial counts remaining below 1.3 log_10_ CFU/mL over 90 days, irrespective of processing and storage conditions. High-pressure processing (HPP) showed promise in extending the shelf-life by reducing the risk of microbial growth. The total polyphenol content remained stable, with no significant impact from processing or temperature. The total antioxidant capacity progressively increased during storage, potentially due to non-oxidative reactions of melanoidins. Although *E. coli* O157:H7, *L. monocytogenes* and *S. enterica* may survive for several weeks on cold brew coffee, the findings of this study confirm that the three main pathogens failed to grow on coffee brews, most likely due to the lack of nutrients or to the presence of naturally occurring compounds with antimicrobial properties. Moreover, it is reported for the first time that high-pressure processing (600 MPa, 3 min) can be used as a nonthermal strategy to ensure food safety and to preserve the quality traits of CBC up to 90 days at 4 °C and room temperature. A >6-log_10_ reduction in vegetative pathogens was observed, and the microbiological, physicochemical and functional qualities of cold brew coffee were maintained. Nevertheless, the impact of coffee variety and roasting levels, extraction conditions and product formulations requires further research to establish the performance of HPP over a wider range of conditions. In addition, the behavior of other microorganisms (such as spore-formers), should be considered for a complete risk assessment of high-pressure-processed cold brew coffee. Limitations were noted in evaluating sensory properties after processing and during storage. Therefore, it is recommended to conduct more research in this area to improve general understanding and address these limitations.

## Figures and Tables

**Figure 1 foods-12-04231-f001:**
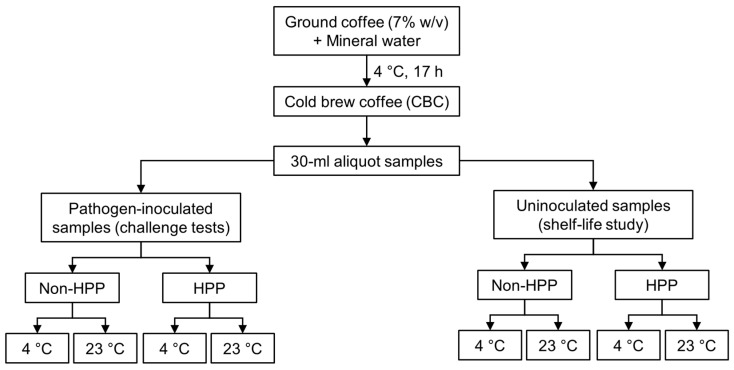
Schematic representation of the experimental design followed for the pathogen-inoculated and uninoculated CBC samples.

**Figure 2 foods-12-04231-f002:**
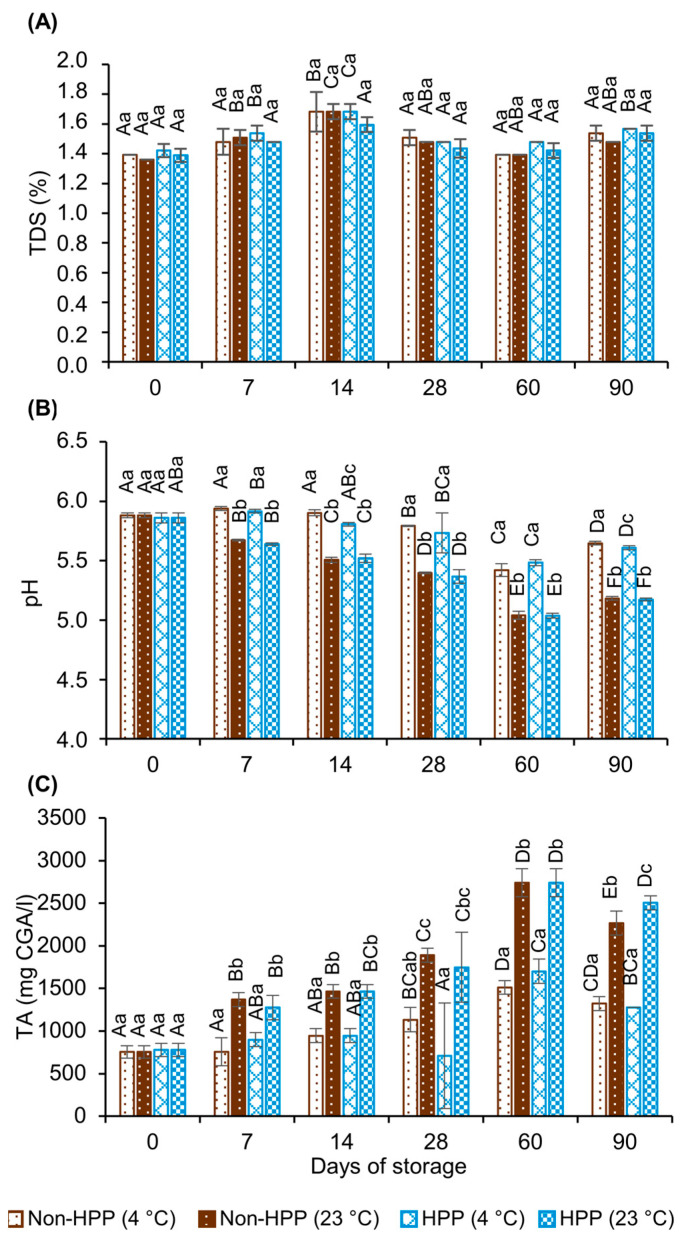
Total dissolved solids (**A**), pH (**B**) and titratable acidity (**C**) of unprocessed and high-pressure-processed (600 MPa, 3 min) cold brew coffee samples stored at 4 °C or 23 °C for 90 days. Results are expressed as the mean ± standard deviation (n = 3). Different uppercase letters indicate significant difference in values for a given processing condition across storage days (*p* ≤ 0.05). Different lowercase letters indicate significant difference in values between processing conditions at the same storage day (*p* ≤ 0.05).

**Figure 3 foods-12-04231-f003:**
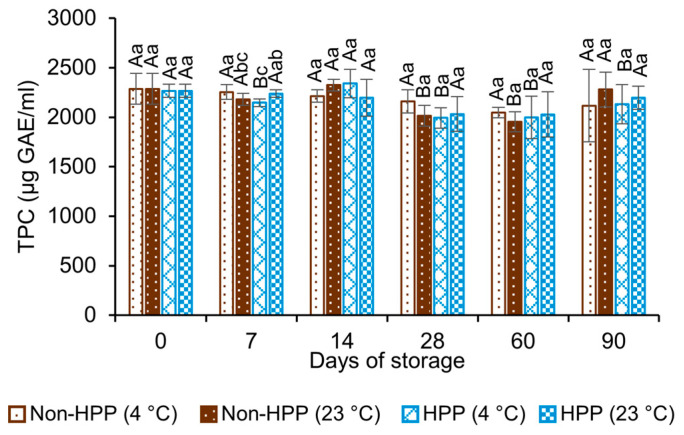
Total phenolic content (TPC) of unprocessed and high-pressure-processed (600 MPa, 3 min) cold brew coffee samples stored at 4 °C or 23 °C for 90 days. Results are expressed as the mean ± standard deviation (n = 3). Different uppercase letters indicate significant difference in values for a given processing condition across storage days (*p* ≤ 0.05). Different lowercase letters indicate significant difference in values between processing conditions at the same storage day (*p* ≤ 0.05).

**Figure 4 foods-12-04231-f004:**
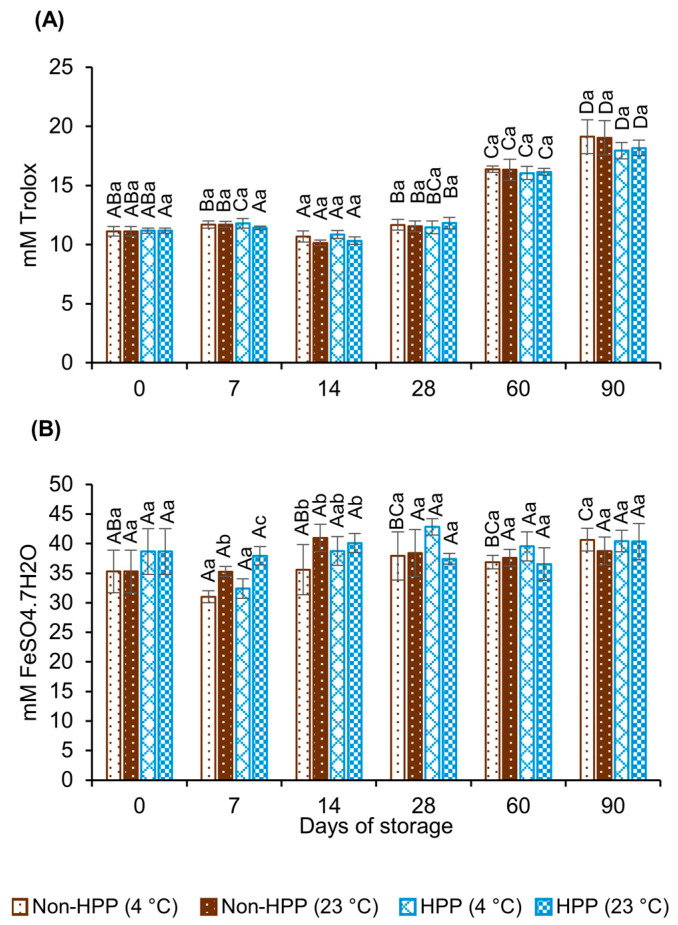
Total antioxidant capacity assessed using ABTS (**A**) and FRAP (**B**) methods of unprocessed and high-pressure-processed (600 MPa, 3 min) cold brew coffee samples stored at 4 °C or 23 °C for 90 days. Results are expressed as the mean ± standard deviation (n = 3). Different uppercase letters indicate significant difference in values for a given processing condition across storage days (*p* ≤ 0.05). Different lowercase letters indicate significant difference in values between processing conditions at the same storage day (*p* ≤ 0.05).

**Figure 5 foods-12-04231-f005:**
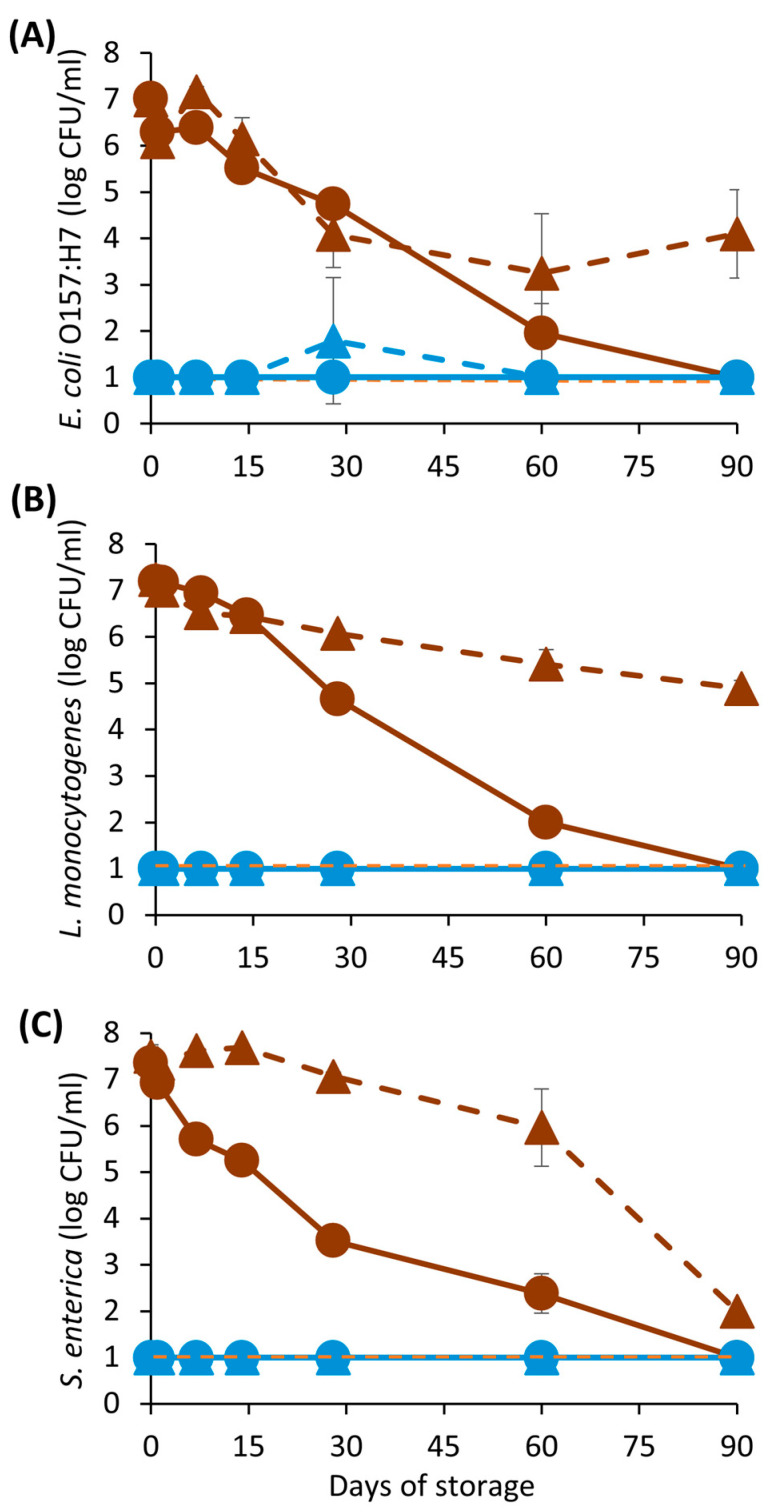
Concentration of *E. coli* O157:H7 (**A**), *L. monocytogenes* (**B**) and *S. enterica* (**C**) on unprocessed (brown) and high-pressure-processed (600 MPa, 3 min) (blue) samples of cold brew coffee stored at 4 °C (circles with continuous lines) or 23 °C (triangles with dashed lines) through 90 days of storage. Results are expressed as the mean ± standard deviation (n = 3). The dotted line represents the limit of detection.

**Table 1 foods-12-04231-t001:** Five-strain bacterial cocktails used for the challenge tests.

Species	Strain	Serotype	Source
*E. coli* O157:H7	CIP ^a^ 106326	O157:H7	Stool, Human, Haemorrhagic colitis outbreak
	CIP 105212	O157:H7	Stool, Human
	CIP 105231	O157:H7	Stool, Child
	CIP 105245	O157:H7	Human, Enteritis case
	CIP 106330	O157:H7	Human
*L. monocytogenes*	FSL ^b^ N3-008	4b	Coleslaw, epidemic, Halifax, 1981
	FSL R2-503	1/2b	Human, epidemic, Illinois, 1994
	FSL J1-094	1/2c	Human, sporadic case
	FSL W1-110	4c	Unknown
	FSL J1-168	4a	Human, sporadic case
*S. enterica*	FSL R8-6671	Dessau	Peanut
	FSL S5-439	Dublin	Human
	FSL S5-373	Braenderup	Human
	HUBU ^c^ 90196	Enteritidis	Human
	HUBU 71144	Typhimurium	Human

^a^ CIP: Pasteur Institute of Paris, France; ^b^ FSL: Cornell University, United States; ^c^ HUBU: Hospital Universitario de Burgos, Spain.

**Table 2 foods-12-04231-t002:** Color coordinates in CIE *L** *a** *b** space of unprocessed and high-pressure-processed (600 MPa, 3 min) cold brew coffee samples stored at 4 °C or 23 °C for 90 days.

Parameter	Day of Storage	Unprocessed		HPP	
		4 °C	23 °C	4 °C	23 °C
*L**	0	3.06 ± 0.72 ^Aa^	3.06 ± 0.72 ^Aa^	2.58 ± 0.27 ^Aa^	2.58 ± 0.27 ^Aa^
	7	4.98 ± 0.56 ^Aa^	5.44 ± 0.56 ^Ba^	5.48 ± 1.13 ^Ba^	5.69 ± 0.69 ^Ba^
	14	2.83 ± 0.42 ^Aa^	3.10 ± 0.35 ^Aa^	2.64 ± 0.27 ^Aa^	3.24 ± 0.37 ^Aa^
	28	2.61 ± 0.50 ^Aa^	2.91 ± 0.57 ^Aca^	2.60 ± 0.35 ^Aa^	2.79 ± 0.69 ^Aa^
	60	5.68 ± 2.65 ^Aa^	6.54 ± 2.21 ^Ba^	3.26 ± 2.21 ^Aa^	6.17 ± 1.26 ^Ba^
	90	3.42 ± 2.94 ^Aa^	1.29 ± 1.43 ^Ca^	2.45 ± 1.37 ^Aa^	2.97 ± 1.19 ^Aa^
*a**	0	1.64 ± 0.57 ^Aa^	1.64 ± 0.57 ^Aa^	1.37 ± 0.45 ^Aa^	1.37 ± 0.45 ^Aa^
	7	−0.71 ± 0.07 ^Ba^	−0.56 ± 0.07 ^Ba^	−0.58 ± 0.11 ^Ba^	−0.52 ± 0.16 ^Ba^
	14	−0.36 ± 0.09 ^CDa^	−0.49 ± 0.15 ^Ba^	−0.36 ± 0.15 ^Ca^	−0.22 ± 0.09 ^BCa^
	28	−0.18 ± 0.11 ^Da^	−0.13 ± 0.29 ^Ba^	−0.36 ± 0.07 ^Ca^	0.13 ± 0.56 ^Ca^
	60	−0.61 ± 0.12 ^BCa^	−0.44 ± 0.42 ^Ba^	−0.54 ± 0.07 ^Ba^	−0.65 ± 0.08 ^Ba^
	90	−0.57 ± 0.32 ^BCa^	−0.48 ± 0.42 ^Ba^	−0.76 ± 0.16 ^Da^	−0.76 ± 0.16 ^Ba^
*b**	0	3.29 ± 0.24 ^Aa^	3.29 ± 0.24 ^Aa^	2.86 ± 0.30 ^Ab^	2.86 ± 0.30 ^ABb^
	7	1.83 ± 0.08 ^Ba^	2.00 ± 0.17 ^BCb^	1.72 ± 0.11 ^Ba^	2.08 ± 0.20 ^CDb^
	14	1.96 ± 0.19 ^Ba^	1.84 ± 0.23 ^Ba^	2.10 ± 0.21 ^Ca^	2.19 ± 0.23 ^Aca^
	28	2.49 ± 0.20 ^Ca^	2.58 ± 0.44 ^Ca^	2.19 ± 0.05 ^Ca^	2.95 ± 0.79 ^Ba^
	60	0.53 ± 0.23 ^Da^	0.79 ± 0.68 ^Da^	0.52 ± 0.60 ^Da^	0.60 ± 0.28 ^Da^
	90	0.57 ± 0.57 ^Da^	0.42 ± 0.67 ^Da^	1.07 ± 0.19 ^Ea^	1.42 ± 0.37 ^Ca^
Δ*E*	0	-	-	0.74 ± 0.30 ^Aa^	0.74 ± 0.30 ^Aa^
	7	3.38 ± 0.28 ^BCa^	3.50 ± 0.37 ^BCa^	3.70 ± 0.85 ^Ca^	3.62 ± 0.60 ^Ca^
	14	2.43 ± 0.16 ^BDa^	2.58 ± 0.24 ^BDa^	2.36 ± 0.19 ^Ca^	2.17 ± 0.14 ^Ca^
	28	2.06 ± 0.25 ^ADa^	1.95 ± 0.46 ^Da^	2.33 ± 0.03 ^Ca^	1.81 ± 0.59 ^Ca^
	60	4.81 ± 1.33 ^Ea^	4.81 ± 1.33 ^Ea^	3.67 ± 0.15 ^Ba^	4.75 ± 0.62 ^Da^
	90	4.24 ± 1.01 ^CDa^	4.02 ± 0.75 ^CEa^	3.46 ± 0.04 ^Ba^	3.09 ± 0.39 ^Ba^

Different uppercase letters between columns indicate significant difference in values for a given processing condition across storage days (*p* ≤ 0.05). Different lowercase letters between rows indicate significant difference in values between processing conditions at the same storage day (*p* ≤ 0.05). The hyphen (-) symbol signifies that the unprocessed sample was utilized as the reference to compute the total color difference (Δ*E*) parameter.

## Data Availability

The data presented in this study are available on request from the corresponding author. The data are not publicly available, due to privacy restrictions.

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
