# Peer review of "High-Pressure Processing for Cold Brew Coffee: Safety and Quality Assessment under Refrigerated and Ambient Storage"

_foods, 2023, doi:10.3390/foods12234231_

Round 1

Reviewer 1 Report

Comments and Suggestions for Authors

Here are some comments and suggestions for your manuscript:

  1. Clarity and Structure:
    • The manuscript appears to be in the middle of a section, and it starts with "The TDS were measured..." It's important to provide some context or introduction to the research at the beginning of the manuscript before diving into the methods section. This will help readers understand the purpose of the study.
  2. Acronyms and Abbreviations:
    • Define acronyms and abbreviations the first time they are used. For example, CBC, HPP, TDS, TA, and TPC should be defined when they are first mentioned.
  3. Methods Section:
    • Provide more information about the study design and objectives in the introduction section before jumping into the methods. What is the main research question or hypothesis you are addressing?
    • Consider breaking the Methods section into subsections for better organization. For example, one subsection for "Sample Preparation," another for "Measurement of Physicochemical Parameters," and so on.
    • Provide more details about the sample size, source of samples, and any randomization or controls used in the study.
    • Clarify the units of measurement for various parameters (e.g., μg GAE/ml) to avoid confusion.
    • Ensure that the methods are described in a logical order. For example, you start with TDS and then move to pH and TA. Consider rearranging them if there is a more logical flow.
    • Explain why specific equipment or methods were chosen for measurements (e.g., why a particular refractometer or pH meter was used).
  4. Results Section:
    • Include a summary or overview of the key findings at the beginning of the Results section.
    • Provide more context for the data presented. For example, explain the significance of changes in TDS, pH, and TA over time and under different conditions.
    • Use tables and figures to present data in a more visually appealing and understandable manner.
    • Avoid excessive use of technical jargon without explanation. Make sure that a general audience can follow the results.
  5. Discussion Section:
    • The discussion section is missing in the provided text. In the discussion, you should interpret your results, compare them to previous research, and discuss the implications of your findings.
  6. Conclusion Section:
    • In the provided text, there's a brief conclusion section. Consider expanding this section to summarize the key findings and their significance in more detail.
Comments on the Quality of English Language

no issues

Author Response

The answers are included in the following document.

Reviewer 2 Report

Comments and Suggestions for Authors

I think this is an interesting topic on how to prepare cold brew coffee. The method is to use high pressure to eliminate common organisms. The procedure is to use a 3-minute treatment and 600 MPa and then wait for some days until the organisms die off. If you do a pasteurization procedure it would kill 100% of the organisms but no longer enable cold brew coffee. The experiment is poorly designed. You are basically studying how the pressure kills microorganisms. You should not use 3 min or treatment but an hour or 10 hours and times in between. You should find out how the strength of the pressure treatment kills the microorganisms. I don't know how long the optimal treatment is for each bacteria. I do know that bacteria can recover from short high-pressure treatment and be pathogenic again. See the research by Pradeep Kumar at Arkansas. I don't think the final claim where you say you can drink the coffee safely is supported. There are still some bacteria left and they were not all killed by the pressure. If you don't want pasteurization you are going to have to find how long of pressure it takes and what magnitude. The experiment you did is good and gives you the tools to try different lengths of pressure treatment. I think you should make this major revision and then resubmit as the methodology is flawed. There is no discussion of why 3 minutes was chosen as a time. It just seems random like something similar to what other papers have.

Author Response

(The authors gave the same response as above.)

Reviewer 3 Report

Comments and Suggestions for Authors

This manuscript deals with the study of High Pressure Processing (HPP) for Cold Brew Coffee. The object is interesting and the treatment method is advanced. While I have few comments for improving readability and emphasizing key results, the manuscript is worth consideration for publication after addressing these comments and revising the manuscript.

1.     As feedstock for the experiments High Pressure Processing was used to Influence on safety of Cold Brew Coffee. One of my initial concerns was whether it would be possible to accurately identify and pinpoint a necessity of HPP in industrial CBC processing. According to the information at Line 31-41 there is no any reference to an industrial processing. Is there a real problem with storage? Which technological step should be adopted for HPP. CBC or water preparation etc. This will make the motivation for this experiment easier to understand in my opinion and introduce the reader better into the topic.

2.     M&M section 2.3. The authors stated: High pressure processing was carried out at 600 MPa for 3 min at ~18 °C in a Hiperbaric 135 liter unit. What is the based for a such parameters. Why 600 MPa? Why 3 min? Any background?

3.     Figure 5. High-pressure processed (600 MPa, 3 min) (blue) samples of cold brew coffee stored at 23 °C (triangles with dashed lines. What happened during 30 day of experiment fig A? Why the triangle of the statistical error was significant?

Based on the above points, I would propose a minor revision of the manuscript.

Author Response

(The authors gave the same response as above.)
